# Identification of Transcriptional Networks Involved in De Novo Organ Formation in Tomato Hypocotyl Explants

**DOI:** 10.3390/ijms232416112

**Published:** 2022-12-17

**Authors:** Eduardo Larriba, Míriam Nicolás-Albujer, Ana Belén Sánchez-García, José Manuel Pérez-Pérez

**Affiliations:** Instituto de Bioingeniería, Universidad Miguel Hernández, 03202 Elche, Spain

**Keywords:** de novo shoot regeneration, de novo root formation, wound-induced cell reprogramming, gene regulatory networks, time-course RNA-Seq

## Abstract

Some of the hormone crosstalk and transcription factors (TFs) involved in wound-induced organ regeneration have been extensively studied in the model plant *Arabidopsis thaliana*. In previous work, we established *Solanum lycopersicum* “Micro-Tom” explants without the addition of exogenous hormones as a model to investigate wound-induced de novo organ formation. The current working model indicates that cell reprogramming and founder cell activation requires spatial and temporal regulation of auxin-to-cytokinin (CK) gradients in the apical and basal regions of the hypocotyl combined with extensive metabolic reprogramming of some cells in the apical region. In this work, we extended our transcriptomic analysis to identify some of the gene regulatory networks involved in wound-induced organ regeneration in tomato. Our results highlight a functional conservation of key TF modules whose function is conserved during de novo organ formation in plants, which will serve as a valuable resource for future studies.

## 1. Introduction

Although plants have an outstanding ability to repair damaged tissues of even regenerate lost organs, our current understanding of the molecular networks involved in their regulation [1,2,3] is mainly limited to studies using the *Arabidopsis thaliana* model system [4]. The first gene regulatory network (GRN) associated with cell reprogramming during plant regeneration was obtained by using an enhanced yeast one-hybrid screen with 252 TFs and 48 gene promoters likely to be relevant for regeneration in *A. thaliana* [5]. Their results revealed several regulatory nodes, including TFs from the APETALA 2/ETHYLENE RESPONSE FACTOR (AP2/ERF) and LATERAL ORGAN BOUNDARY/ASYMMETRIC LEAVES 2 DOMAIN (LBD) families, which might have overlapping targets associated with responses to stress, cytokinin, auxin and cell proliferation [5]. More recently, a web-based application and analytical platform, REGENOMICS, was implemented and contains all currently available regeneration-specific RNA-sequencing (RNA-Seq) datasets. REGENOMICS allows users to perform integrative analyses to generate co-expression networks and provide functional information about plant regeneration processes [6]. However, the current version of REGENOMICS is composed primarily of *A. thaliana* RNA-Seq datasets, making it difficult to find conserved factors involved in plant regeneration through multi-species analyses.

Several studies have identified various wound-induced TFs that have critical roles in regeneration [1,2,3]. One of these TFs belonging to the AP2/ERF family is WOUND INDUCED DEDIFFERENTIATION 1 (WIND1) and its closest homologs WIND2–WIND4. WIND TFs promote callus formation by activating CK response [7], and their ectopic expression can bypass both wounding and auxin pre-treatment, as well as increase de novo shoot regeneration from root explants of different plant species, including tomato [8,9]. WIND1 promotes shoot regeneration through the direct activation of the AP2/ERF TF ENHANCER OF SHOOT REGENERATION 1 (ESR1) [10]. Recent results from time-course transcriptome analysis upon *WIND1* induction have revealed that WIND TFs function as key regulators of wound-induced responses by promoting dynamic transcriptional alterations of genes involved in cell reprogramming, vascular regeneration and defense responses [11].

In this study, we carried out a bioinformatics approach to characterize the expression of all TF genes during de novo organ formation in *Solanum lycopersicum* “Micro-Tom” using the well-established model of young hypocotyl explants developed in our laboratory [12,13,14]. We have annotated 1399 expressed genes putatively encoding TFs and studied their expression profiles using k-means [15] and STEM [16] analysis in the apical and basal regions of the hypocotyl and over time. Because known TF modules required for organ regeneration are conserved between distant plant species, we have established homologous GRNs for some of the TF genes that showed transient upregulation during cell reprogramming and/or tissue specification. New experimental data on cell cycle regulation and reactive oxygen species (ROS) distribution are also provided. Our results have revealed candidate TFs to regulate tissue regeneration that could serve to bridge the gap in our knowledge about tissue-specific wound-induced organ formation in tomato.

## 2. Results

### 2.1. Expression of Genes Encoding TFs during De Novo Organ Formation

From a previous RNA-Seq experiment [13], we identified 1399 expressed genes putatively encoding TFs that were assigned to 48 families (Appendix A). We found that 125 of these genes (8.93%) were preferentially expressed in the apical region of hypocotyl explants between 0 and 192 h after excision (hae; Figure 1a), and 33 of them were specifically expressed in the apical region over time. We constructed a TF interaction network based on the results of STRING analyses using data from its *A. thaliana* orthologs and found significantly enriched gene ontologies (GOs) related to shoot meristem formation and leaf development (Figure 1b). Tomato orthologs of *WUSCHEL* (*WUS*; Solyc02g083950), *AGAMOUS* (*AG*; Solyc02g071730) and *CUP-SHAPED COTYLEDON 2* (*CUC2*; Solyc07g062840), among others, were upregulated at specific time points in the apical region (Figure 1c and Appendix A). On the other hand, 271 TF genes (19.37%) were preferentially expressed in the basal region of hypocotyl explants over time, and 101 of these genes were exclusively expressed in the basal region (Figure 1d and Appendix A). Through STRING analysis, we identified genes belonging to the TF families of LBD and PLETHORA/AINTEGUMENTA-LIKE (PLT/AIL); some of its members are well-known regulators of the specification and growth of the root meristem (Figure 1e). Tomato orthologs of *WUSCHEL-RELATED HOMEOBOX 5* (*WOX5*; Solyc03g096300), *PLT1/2* (Solyc11g061750), *PUCHI* (Solyc01g067540), *FEZ* (Solyc05g007550) and *SOMBRERO* (*SMB*; Solyc12g017400) were upregulated during adventitious root (AR) formation in the basal region of hypocotyl explants. In addition, the expression of the superlocus *SHOOTBORNE ROOTLESS* (*SBRL*; Solyc09g066260 and Solyc09g066270), which controls the initiation of ARs and lateral roots (LRs) in tomato [17], showed higher expression in the basal region of hypocotyl explants after wounding (Figure 1f). These results confirm the functional relevance of our transcriptome dataset for identifying the transcriptional networks involved in de novo organ formation in tomato.

### 2.2. Analysis of TF Expression Profiles during De Novo Organ Formation

The 1296 TF genes that were expressed in the apical region of the hypocotyl were grouped into eight expression clusters. Clusters C1 to C4 included TF genes whose expression decreased over time, whereas TF gene expression in clusters C5 to C8 increased relative to excision time (Figure 2a). To investigate the expression profiles of TF genes, we performed a time-series analysis using STEM [16]. Ten different gene expression profiles were chosen for further study. Most of the constitutively downregulated TF genes were found in P0, P2, and P7 expression profiles, while P3 included specifically downregulated TF genes at 24–96 hae (Figure 2b,c). On the other hand, constitutively upregulated TFs were distributed across P13, P14 and P15 expression profiles (Figure 2b,d). Most of the upregulated TF genes at 24 hae were included in the P11 profile, whereas the P5 and P6 profiles included upregulated TF genes at the final time point of 192 hae (Figure 2b,e). Next, we selected 111 TF genes, whose orthologs in *A. thaliana* have previously been linked to regeneration (Appendix A), to use them as markers for the characterization of the studied expression profiles (Figure 2c–e). In *A. thaliana*, WIND1 and ESR1 establish a central regulatory hub necessary for shoot regeneration [10]. Consistent with their role in shoot regeneration, tomato orthologs of *WIND1* (Solyc04g072900) and *ESR1/2* (Solyc05g013540) were upregulated at 24 hae in the apical region of the hypocotyl and were recovered from P7 and P14 profiles, respectively (Figure 2c,d). Tomato orthologs of know genes required for root development, such as *SCARECROW* (*SCR*; Solyc10g074680) or *PLT5* (Solyc07g018290), were found to be downregulated in the apical region of hypocotyl explants during de novo shoot formation and were included in the P0 profile (Figure 2c,f).

Similar to what was found in the apical region, k-means analysis of the 1364 TF genes expressed in the basal region of the hypocotyl also yielded eight expression clusters (Figure 3a). Most of the downregulated TFs in clusters C1 to C3 were assigned by STEM analysis to gene expression profiles P0, P2, P3, and P7 (Figure 3b,c). In contrast, the upregulated TF genes in the basal region of the hypocotyl during AR formation (clusters C6 to C8) were assigned to P13, P14, and P15 expression profiles (Figure 3b,d). TF genes included in the P5 and P6 profiles also displayed dynamic expression patterns, which were characterized by downregulation at earlier time points and upregulation at later time points (Figure 3b,e). Most of these TF genes were found in the C4, C7 and C8 k-means clusters (Figure 3b). The C5 cluster contained upregulated TF genes at 24 hae, and almost all were included in the P11 profile (Figure 3b,e). Some previously known TF genes involved in cell reprogramming and plant regeneration (see above) were assigned to the studied STEM profiles (Figure 3c–e). Many know regulators of shoot regeneration, such as *WIND1* and *ESR1*, were downregulated in the basal region of the hypocotyl (Figure 3c), in opposite contrast to that found for these genes in the apical region (see above). On the other hand, the expression of putative tomato orthologs of some key regulators of root specification and growth, such as *PLT/AIL* genes (*PLT1/2*, *PLT3* (Solyc11g010710), and *BABY BOOM* (*BBM*; Solyc11g008560)), *MONOPTEROS* (*MP*; Solyc04g081240) and *WOX8/9* (Solyc02g077390), among others, were upregulated over time specifically in the basal region of the hypocotyl (Figure 3f). *PLT5* and *SHORT-ROOT* (*SHR*; Solyc02g092370) expression decreased from 24 to 96 h, and their steady-state levels recovered at 192 h, coinciding with AR emergence (Figure 3c,f). Among the genes deserving of further study is the putative tomato ortholog of *WOX13*, which was found in the P11 expression profile (Figure 3e,f). Taken together, our results have enabled us to identify putative TFs that might be involved in de novo AR and shoot formation in tomato hypocotyl explants after wounding.

### 2.3. GRNs Required for De Novo Organ Formation

In our previous work [13,14], we identified some GO categories that were specifically enriched in the apical or basal regions of the hypocotyl and at different times during de novo organ formation. To investigate the regulatory networks of TFs and their targets that contribute to the observed differences between de novo AR and shoot formation, we focused on specific STEM expression profiles. We searched for putative TF gene targets in each profile and performed GO ontology enrichment analyses between them (see Section 4). We reasoned that TF genes upregulated at 24 hae, most of which are included in the P11 profile in both regions (Figure 2e and Figure 3e), could be important for cell reprogramming and tissue specification during de novo organ formation.

In the apical region, we generated GRNs that included TFs of the P11 and P13 profiles and their putative target genes that were differentially expressed over time (see Section 4). Among the upregulated targets of TFs expressed in the P11 profile, we found GO enrichment for terms related to cell cycle regulation and glycolate/glyoxylate metabolism (i.e., photorespiration), while the downregulated targets associated with TF genes in this profile were enriched for GO terms related to the light-harvesting complex of photosynthesis and auxin responses (Appendix A). We found that putative tomato orthologs of *ERF3* (Solyc07g064890), *BASIC LEUCINE-ZIPPER 44* (*bZIP44*; Solyc01g109880), and *ELONGATED HYPOCOTYL 5* (*HY5*; Solyc08g061130) connected different sets of core cell cycle regulators (Figure 4a,c). Most of the cell cycle targets of bZIP44 and HY5 overlapped (Figure 4a and Appendix A), suggesting a cooperative function of these two TFs. In contrast, the Solyc03g116100 node, which encodes the putative tomato ortholog of MYB DOMAIN PROTEIN 94 (MYB94) and MYB96, connected some genes in the auxin pathway that were downregulated over time in the apical region (Figure 4a,d).

The P13 profile of constitutively upregulated TF genes in the apical region included TFs involved in shoot apical meristem specification, such as *WUS* (Figure 2d). We found that GO terms related to cell cycle regulation and oxidoreductase activity were specifically enriched among upregulated targets of TF genes assigned to this profile (Appendix A). One of the most prevalent cell cycle-related TF nodes in this profile was that of Solyc03g113760 (Figure 4b), which putatively encodes the DP-E2F-LIKE 1 (DEL1) protein known to inhibit the endocycle in *A. thaliana* [18]. Two other TF nodes, Solyc01g095460 and Solyc06g065820 (Figure 4e), encoding the putative orthologs of G-BOX BINDING FACTOR 3 and ETHYLENE AND SALT INDUCIBLE 3, included genes that might be involved in ROS regulation (Figure 4b,f). We found 25 putative targets involved in cell cycle regulation that were included in the GRNs of P11 and P13 profiles, most of which are involved in DNA replication and showed upregulation at 24 and 192 hae (Appendix A).

In the basal region, most of the upregulated targets of TF genes within P11 expression profile (Figure 3e) were related to ribosome function, whereas the downregulated target genes were enriched for photosynthesis-related and oxidoreductase activity GO terms (Appendix A). Of the many GRNs entangled in this profile (Appendix A), we focused on those of Solyc01g067540 and Solyc05g054400, putatively encoding orthologs of *PUCHI* [19] and *GATA15/17* [20], respectively, in *A. thaliana*. The node of GATA15/17 included genes encoding glutaredoxins that were deregulated over time, whereas the node of PUCHI included several genes encoding constituent proteins of the cytoplasmic ribosome (Figure 5a and Appendix A). Some of the key regulators of root founder cell specification, such as SBRL or LBD16 [21] were assigned to the P15 profile (Figure 3d). From this profile, a complex GRN involving TF genes and their putative targets was constructed (Appendix A). The TF gene with the highest number of interactions with its putative deregulated targets was Solyc11g008560 (Figure 5b), which encodes the BBM TF that is required in cooperation with other *PLT* genes to maintain stem cell activity in *A. thaliana* roots [22]. Among the putative BBM targets identified, several cell cycle-related genes encoding cyclins (e.g., Solyc11g005090 (*CycA1;1*), Solyc02g082820 (*CycB2;1/CycB2;2*) and Solyc02g092980 (*CycD3;3/CycD3;2*)), were found to be upregulated after 24 h in the basal region of the hypocotyl (Figure 5c). Some BBM targets involved in cell wall biogenesis (e.g., cellulose synthases and pectin methylesterases) were also similarly regulated (Figure 5c). On the other hand, putative BBM targets involved in the transduction of various hormones, such as auxins (*SlARF1*, *SlIAA14*, *SlIAA17* and *SlIAA7*) or abscisic acid, were strongly downregulated starting at 24 h (Figure 5d). We found several BBM targets encoding ribosomal proteins, which do not overlap with PUCHI targets encoding other ribosomal proteins (Appendix A).

### 2.4. The Cell Cycle Is Differentially Regulated in the Apical and Basal Regions during de Novo Organ Formation

Since the results found above suggested a differential regulation of the cell cycle between the apical and basal regions of the hypocotyl, we measured the DNA content by flow cytometry (Figure 6a,b) and found significant differences in the ploidy profiles of the apical and the basal region of the hypocotyl over time (Figure 6c and Appendix A). We estimated the mitotic index (MI) and endoreduplication index (EI) as previously described [23]. In the apical region, MI increased significantly at 24 and 96 hae relative to 0 hae, and MI values were restored to steady-state levels at 192 hae (Figure 6d). In the basal region at 0 hae, MI values were higher than those in the apical region and did not change significantly (Figure 6d). The apical region at 0 hae showed the lowest EI values, which were similar in both regions and over time (Figure 6e).

We found 172 expressed genes that are associated with core cell cycle functions, 97 of which were deregulated over time (Appendix A). We classified these genes into 16 functional groups and performed an enrichment analysis with respect to their expression at 0 hae (Appendix A). In the apical region, we found an enrichment of upregulated DEGs related to DNA replication, DNA repair, and genes involved in G1/S transition or mitotic regulation, as well as downregulated DEGs involved in inhibition of endoreduplication (Figure 6f). On the other hand, upregulated DEGs related to cell proliferation in the basal region were found to be enriched at 96 hae (Figure 6g). Some genes encoding central regulators of mitosis, such as B-type CDKs (e.g., Solyc04g082840 and Solyc10g074720), genes encoding regulatory subunits of the anaphase promoting complex or cyclosome (APC/C) (e.g., Solyc03g096870 and Solyc06g043150), or genes associated with the mitotic spindle checkpoint (e.g., Solyc03g007060 and Solyc08g066050), were expressed at different time points in the apical and basal regions (Appendix A and Figure 6g). Most genes encoding A-type and B-type mitotic cyclins were upregulated in the apical and basal regions after wounding, but at different times between regions. The Solyc04g078310 and Solyc12g088530 genes, encoding the CycA3 cyclins whose *A. thaliana* orthologs are required for cell cycle transitions [24], were specifically upregulated at 24 hae in the apical region (Figure 6f). The *CycA* and *CycB* genes were strongly downregulated in the basal region at 24 hae and increased their expression at later times (Figure 6g). In turn, the expression of Solyc07g052610, which encodes an atypical cyclin [25], was differentially upregulated in the apical and basal regions of the hypocotyl explants after wounding (Appendix A).

### 2.5. Characterization of ROS Accumulation during De Novo Organ Formation

Previous results suggested the involvement of ROS homeostasis in wound-induced regeneration in tomato hypocotyls [14]. To assess ROS production, we visualized H_2_O_2_ accumulation during wound-induced organ formation by staining with 3,3’-diaminobenzidine (DAB). At 24 hae, an increase in DAB staining was observed in the apical region of the explants (Figure 7a). The DAB precipitate was confined to tissues near the wounding site at 96 hae, in both the apical and basal regions of the explants (Figure 7a, 96 hae). At 192 hae, DAB staining decreased substantially and was restricted to more distal cells in the basal region and to specific cells within the apical callus (Figure 7a, T8). We confirmed the DAB-specific staining of H_2_O_2_ by incubation with catalase (Figure 7a, 24 hae cat).

Due to these specific patterns of ROS accumulation, we identified 110 DEGs related to ROS homeostasis (Figure 7b and Appendix A) that were assigned to different families based on the enzymatic activities of their products [26] (Appendix A). In the apical region, the expression of most genes encoding ROS-scavenging proteins was upregulated (Appendix A). Furthermore, the expression of five genes encoding membrane-dependent NADPH oxidases of the RBOH family (Appendix A), which are involved in stress-induced singlet oxygen production [27,28], was upregulated in both the apical and basal regions (Appendix A). Since an excess of singlet oxygen could cause increased ROS levels eventually leading to cell death [29], we investigated cell death by trypan blue staining (Figure 7c). At 24 hae, we observed faint trypan blue staining in the distal end of the apical region and no trypan blue staining in the basal region (Figure 7c, 24 hae). However, at 96 hae, trypan blue stained the outer cell layer of the basal region and the more distal cell layer of the apical region of the explants (Figure 7c, 96 hae). Similar trypan blue staining was observed in distal cells within the apical and basal regions of the hypocotyl explants at 192 hae (Figure 7c, 192 hae), indicating that the degree of ROS production was not associated directly with cell death. Genes encoding peroxiredoxins (PRX), thioredoxins (TRX), ascorbate peroxidases (APX), and catalases (CAT) were differentially regulated between the apical and basal regions (Appendix A), which could explain the differences observed in the accumulation of H_2_O_2_ after wounding (Figure 7a).

We identified 85 expressed genes encoding class III peroxidases (POX), 81 of which (95.3%) were deregulated to a greater degree than genes encoding other ROS-scavenging proteins (Appendix A). We grouped these genes according to six different expression profiles (Figure 7d–g). Forty-one DEGs were constitutively upregulated in both the apical region and basal regions over time (e.g., *Prx54*, *Prx68*), and some of these genes (e.g., *Prx92*, *Prx97*) were downregulated in the basal region at 24 hae (clusters A and B, respectively; Figure 7d). Twenty-nine DEGs showed contrasting deregulation in the apical and basal regions, and 21 of these DEGs were upregulated primarily in the basal region (e.g., *Prx100*, *Prx105*, clusters C and D; Figure 7e), and eight DEGs were upregulated only in the apical region (e.g., *Prx44*, *Prx95*, cluster E; Figure 7f). Eleven DEGs were mainly downregulated in both tissues (e.g., *Prx20*, *Prx87*, cluster F; Figure 7g). Our results indicate a differential regulation of POX-encoding genes in the apical and basal regions of hypocotyl explants over time, which could explain the dynamic ROS homeostasis observed during wound-induced organ formation in tomato hypocotyl explants.

### 2.6. CDF3 Is Required for De Novo Shoot Formation in Tomato Hypocotyl Explants

Our results indicate a differential regulation of the GRNs involved in de novo organ formation in the apical and basal region of the hypocotyl. To search for TFs putatively involved in tissue-specific cell reprogramming, we selected TFs with contrasting expression levels whose orthologs are known to regulate environmentally regulated developmental traits. One of these TFs was CYCLING DOF FACTOR 3 (CDF3), encoded by Solyc03g115940, which was identified in two contrasting expression profiles over time: the P6 profile in the apical region (Figure 2e) and the P14 profile in the basal region (Figure 3d). Using PlantRegMap and our RNA-Seq dataset (see Section 4), we identified 1063 deregulated genes (773 and 757 DEGs in the apical and basal regions of the hypocotyl, respectively; Appendix A) among previously described putative CDF3 targets [14]. Our GO enrichment analysis suggests that CDF3 regulates auxin-related genes and other TF genes (Appendix A). Some of the TF genes that might be downregulated by CDF3 in the apical region of hypocotyl explants are Solyc12g010800, the tomato ortholog of *AGL15*, and Solyc01g096070, which putatively encode the ortholog of *ARF11* and *ARF18* in *A. thaliana* (Appendix A). Regarding the basal region of the hypocotyl, we found enrichment in GO terms related to inorganic anion transport (Appendix A).

To characterize the relevance of this TF node during de novo organ formation, we studied two overexpression (OE) lines (#5.2 and #11.2) and one RNAi line (#1.1) of *CDF3*, which were compared with the Moneymaker (MM) background (see Section 4). Although the RNAi line showed a slight delay in germination compared to the other lines, we found no significant difference in the number of LRs after root tip excision with respect to their genotype (*p* value = 0.486; Appendix A). After whole root excision, AR emergence from the hypocotyl occurred slightly earlier in the OE lines than in the RNAi line and MM (*p* value < 0.05; Figure 8a). However, the RNAi line showed a significant increase (*p* value < 0.01) in the number of ARs at 5 days after excision (dae) compared to the *CDF3* OE lines and MM (Figure 8b). At 10 dae, we excised the shoot apex and studied de novo shoot regeneration as previously described [14]. The RNAi line showed a slight but significant delay (*p* value < 0.01) in de novo shoot regeneration compared to the OE lines (Figure 8c), which resulted in a smaller root system of the RNAi line at the end of the experiment (Figure 8d). These results suggested that CDF3 levels should be tightly regulated during de novo organ formation in a tissue-specific context, acting as a negative regulator of AR initiation but as a positive regulator of de novo shoot formation. Due to the known redundancy among the *CDF* genes [30], further studies will be needed to determine the relevance of this DNA binding with one finger (DOF) subfamily of TFs in wound-induced cell reprogramming in tomato.

## 3. Discussion

In previous work from our laboratory, we aimed to study de novo organ formation in tomato using “Micro-Tom” hypocotyl explants after wounding [12]. The current working model involves spatial and temporal regulation of auxin-to-CK gradients in the apical and basal regions of the hypocotyl soon after wounding to regulate cell reprogramming and founder cell activation [14]. We also found that wounding induced extensive metabolic reprogramming of some cells in the apical region of the hypocotyls, highlighting a key role of photorespiration in supplying the energy and structural elements necessary for initial callus growth preceding de novo shoot formation [13]. In this work, we extended our RNA-Seq analysis to identify some of the GRNs involved in wound-induced de novo organ formation in this species.

We found that the expression of the tomato *WUS* homologue (Solyc02g083950) in the apical region of the hypocotyl after wounding defined a GRN that could be involved in the specification of the new shoots. Previous studies in tomato have shown that after laser ablation of the central zone of the shoot apical meristem, *WUS* was ectopically induced in neighboring cells some distance from the lesion, which anticipated the establishment of new regenerated shoots [31]. Furthermore, Solyc02g071730 and Solyc05g009380, putatively encoding AG and JAGGED (JAG) TFs, showed a *WUS*-like expression profile, and we hypothesize that they could contribute to de novo shoot formation. A weak gain-of-function *WUS* allele in tomato, termed *locule number*, affected a CArG element downstream of *WUS* coding region that is capable of binding to AG and downregulating *WUS* expression that, in turn, terminates meristem activity during flower development [32]. *JAG* loss-of-function alleles in tomato (i.e., *lyrate* mutants) caused a broad reduction in overall leaf growth, but increased leaflet dissection by interacting with both the *KNOTTED-LIKE HOMEOBOX* (*KNOX*) genes and the auxin transcriptional network [33]. We hypothesize that in tomato, similar to what was found in *A. thaliana* [34], JAG could antagonize meristem identity genes, such as *WUS*, in cells that are switching from meristem to organ primordium identity, and repressed the *KIP RELATED PROTEIN* (*KRP*) genes in these cells, thus releasing the constraint for DNA replication, a hypothesis than can now be experimentally validated.

K-means and STEM analysis allowed us to identify TF genes with unique expression patterns during de novo organ formation in tomato. We found putative orthologs of two conserved TF modules in *A. thaliana* that have been described as having key roles in wound-induced cell reprogramming and cell cycle reentry during shoot regeneration. On the one hand, Solyc04g072900 and Solyc05g013540, which encode the tomato orthologs of *WIND1* and *ESR1/2*, respectively, were upregulated at 24 hae in the apical region. WIND1 promotes callus formation and shoot regeneration by activating *ESR1* expression [10]. Our results suggest that the function of the WIND1-ESR1/2 module during shoot regeneration might be conserved beyond the Brassicaceae, although other experiments with known loss-of-function tomato *ESR1/2* mutants (i.e., *leafless*) [35] wait for confirmation. Other conserved module included Solyc07g042260 (*AUXIN RESPONSE FACTOR 7* (*ARF7*)/*ARF19*) and Solyc11g071300 (*MYB DOMAIN PROTEIN 3R-4* (*MYB3R4*)), which in *A. thaliana* has been recently shown to regulate cell cycle activation of leaf mesophyll protoplasts in response to exogenous auxin [36]. Implicit GRNs were constructed with the target genes from the upregulated TF genes in selected STEM profiles in the apical region (P11 and P13). One such upregulated node was that of Solyc03g116100, which encodes the putative tomato ortholog of TFs MYB94 and MYB96 connecting some downregulated genes in the auxin pathway (*PIN-FORMED 1*, *ARF4*, etc.) at earlier time points during de novo shoot formation. In *A. thaliana*, MYB94 and MYB96 repress *LBD29* expression, thus playing negative regulatory roles in auxin-induced and callus formation mediated by ARF7 and ARF19 TFs [37]. Therefore, our results suggest a functional conservation of key TF modules involved in the regulation of wound-induced de novo shoot formation, which need to be confirmed experimentally using available genetic tools (known mutant alleles and CRISPR/Cas9-mediated gene editing). Functional analysis of CDF3 in *A. thaliana* suggests a direct role for this DOF TF in drought and osmotic stress responses, likely through regulation of cell integrity, carbon and nitrogen metabolism, and ROS homeostasis [38]. The ectopic expression of CDF3 in tomato increases the photosynthetic rate and biomass production under salinity stress by facilitating nitrogen assimilation and sucrose availability [30,39]. We found that *CDF3* RNAi lines showed a slight delay in de novo shoot regeneration from tomato hypocotyl explants, suggesting that it might also play a role during cell reprogramming. Our previous results [13] indicate that apical tissues in hypocotyl explants need to be reprogrammed into sink tissues; thus, CDF3 may be required for this process. Alternatively, CDF3 could indirectly regulate the auxin-to-CK gradient in apical tissues during de novo shoot regeneration [14] through its auxin-related TF targets.

Several TF genes whose putative orthologs are required for root development in *A. thaliana* (*PLT1/2*, *BBM, FEZ* and *SMB*) were specifically expressed in the basal region of the hypocotyl during AR formation, suggesting their functional conservation in wound-induced AR formation in tomato. Instead, two *PLT* genes, *PLT3* and *PLT7*, were expressed in the apical and basal regions of the hypocotyl after 24 hae, which is consistent with their positive role in establishing pluripotency during de novo organ formation, as was previously shown in *A. thaliana* [40]. We also recovered the expression of the two *LBD* genes from the *SBRL* superlocus, Solyc09g066260 and Solyc09g066270, in the basal region of the hypocotyl [17], confirming that *SBRL* expression can be used as an early marker for AR specification. Two *WOX* genes, Solyc03g096300 and Solyc02g077390, encoding the putative tomato orthologs of *WOX5* and *WOX8/9*, respectively, were also recovered as early markers for wound-induced AR formation in our study. Recent work identified a CRISPR/Cas9 *cis*-regulatory allelic series of the tomato *WOX8/9* gene that exhibited alterations in inflorescence development, whereas *wox8/9* null mutants are embryonically lethal, as in *A. thaliana* [40]. To confirm the functional role of WOX8/9 expression in wound-induced AR formation in tomato, a phenotypic analysis of the available *wox8/9* promoter mutants is needed.

One of the few genes specifically upregulated in the basal region of the hypocotyl at 24 hae was *PUCHI* (Solyc01g067540), whose expression in *A. thaliana* is directly regulated by LBD16 in LR founder cells, downstream of the SLR/IAA14-ARF7-ARF19 module [19]. PUCHI is required for the correct timing of meristem establishment by repressing the premature expression of key meristematic genes in the early stages of LR development, such as *PLT1*, *BBM*, and *WOX5* [41]. *PLT1/2* (Solyc11g061750) and *BBM* (Solyc11g008560) were specifically expressed in the basal region of the hypocotyl explants after 24 hae. The upregulation of PLT1/2 and BBM was correlated with the downregulation of some of their Aux/IAA target genes, such as Solyc06g008590 (*SlIAA17*), Solyc06g053830 (*SlIAA7*) and Solyc09g083290 (*SlIAA14*), which are the putative orthologs of *SLR/IAA14* required for specification of LR founder cells in *A. thaliana* [21]. These results suggest that the specification of AR founder cells in the basal region of tomato hypocotyls after wounding is based on the auxin-mediated SBRL-PUCHI module, while subsequent AR initiation occurs through local activation of the PLT1/2-BBM-WOX5 module. Our results also suggest a negative feedback loop involving PLT1/2-BBM that restricts the specification of AR founder cells in the basal region of the hypocotyl at later time points. Two tomato genes encoding ERF105 (Solyc03g093560) and ERF109 (Solyc10g050970) TFs were specifically expressed in the basal region of the hypocotyl and were strongly downregulated after wounding. While ERF105 has been linked to regulate cold acclimation responses [42], transcriptomic analysis of excised leaves in *A. thaliana* identified ERF109 as a direct target of jasmonic acid (JA) signaling to establish a transient pulse of auxin production necessary for cell cycle activation and de novo root formation [43,44]. These results are consistent with our findings of a steep methyl-JA gradient in the basal region of the hypocotyl soon after wounding (0-10 hae) that anticipated AR formation [14] and that could explain the observed downregulation of these two ERF TFs.

Our expression profiling analysis allowed us to identify commonly deregulated TF genes in the apical and basal regions of the hypocotyl, which might be involved in early events (i.e., cell dedifferentiation) during de novo organ formation. One of these regulators is Solyc02g082670 (*WOX13*), since its expression increased significantly at 24 hae in both tissues. A recent report identified WOX13 as a key regulator of callus formation in *A. thaliana*, downstream of auxin signaling and WIND1 [45,46]. A previous study showed that WOX13 orthologues in the moss *Physcomitrium patens* are involved in cellular reprogramming at wounding sites [47]. Parallels in WOX13 function between plant species suggest the existence of a conserved GRN involving WOX13 that regulates cell reprogramming after tissue damage. On the other hand, Solyc08g077110, which encodes the NAC DOMAIN CONTAINING PROTEIN 71 (NAC071) and NAC096 homologs in *A. thaliana*, was also upregulated after wounding in the apical and basal regions of the hypocotyl. In *A. thaliana*, *NAC071* and *NAC096* are induced by endogenous auxin in different regions near the wounding in cut stems and act redundantly to promote cell proliferation of cambial cells during tissue reunion [48,49]. Another key regulator of cambial cell division during de novo organ formation could be Solyc11g068960, which putatively encodes the *LONESOME HIGHWAY* (*LHW*) TF, which in *A. thaliana*, regulates formative divisions in vascular cells in roots, leaves, and stems, downstream of auxin [50,51]. Therefore, our transcriptomic analysis and implicit GRN reconstruction could provide candidate TF genes whose function is conserved during de novo organ formation in plants.

We found significant differences in the ploidy profile of cells in the basal and apical regions of the hypocotyl at the time of wounding, suggesting region-specific cell cycle regulation. Our results suggested that basal tissues contained a higher proportion of mitotic and endoreduplicated cells than apical ones at 0 hae, and that a small increase in these two cell populations occurred in the basal region of the hypocotyl during de novo root formation. On the other hand, the apical region was characterized by a strong regulation of the cell cycle after wounding, which is necessary for callus formation during de novo shoot initiation. MI and EI were significantly increased at 24 hae in the apical region of the hypocotyl, which coincided with the upregulation of key genes involved in DNA replication and mitotic regulation, such as those of *CycA3* (Solyc04g078310 and Solyc12g088530) and *CycP2;1* (Solyc07g052610) cyclins, whose orthologs in *A. thaliana* are respectively required to control formative cell divisions [52] and the G2/M transition of meristematic cells in response to sugar [25]. Two TF genes, Solyc04g081350 (*E2FC*) and Solyc03g113760 (*E2FE*, also known as *DEL1*), were specifically deregulated in the apical region of the hypocotyl over time (profiles P11 and P13). It has been proposed that E2FC acts as a transcriptional repressor necessary for the switch between cell division and endoreduplication, which could be important in establishing the developmental gradient of cell divisions in growing organs [53]. On the other hand, DEL1 preserves the mitotic state of proliferating cells by suppressing transcription of genes necessary to enter the endocycle [18]. Our GRN analysis identified a DEL1 regulatory node that includes upregulated target genes encoding key components of the replication licensing factor complex, such as ORC6 (Solyc05g007450), MCM4 (Solyc01g110130), MCM6 (Solyc08g082200) and MCM10 (Solyc01g111550), and a DNA primase (Solyc08g082200), among others. These results revealed an unprecedented role of tomato DEL1 TF in the regulation of G1/S transition during de novo organ formation, which deserves further attention.

In our previous study [13], we observed differential regulation of photosynthesis-related genes in the apical and basal regions of hypocotyls after wounding that resembled adaptation to high-light stress. We observed a local increase in ROS production in the apical region at 24 hae, which decreased at the later time point, probably due to enhanced expression of genes encoding PRX and TRX, which are required for chloroplast performance under stress conditions [54]. In this study, we found striking patterns of ROS accumulation in the apical and basal regions of hypocotyl explants over the studied time course, which were not directly correlated with the degree of cell death and therefore might have a signaling role during de novo organ formation. The accumulation of ROS in the basal region of the hypocotyl explants at 96 and 192 hae could be related to AR emergence [55]. During primary root and LR development in *A. thaliana*, ROS distribution (which is regulated by specific POX enzymes) is required for the transition between cell proliferation and differentiation within the meristem [56,57]. POX performs different functions in addition to its primary role in wound response [58,59]. Some of the POX-encoding genes were strongly deregulated in our RNA-Seq dataset. In *A. thaliana*, a subset of class III *POX* genes was targeted by LBD29 [60], a master regulator of auxin-induced callus formation [61]. Furthermore, the biological responses regulated by LBD29 closely resemble those regulated by hypoxia treatment, which has long been considered a crucial factor in determining cell fate in mammals [60]. We found several *POX* genes (*Prx05*, *Prx11*, *Prx25*, *Prx26*, *Prx38* and *Prx68*) whose expression was upregulated in the apical region of hypocotyl explants and that were structurally related to the LBD targets described above, suggesting that a GRN involving LBD29-like TF might be conserved in tomato. In addition, the natural variation in hormone-induced shoot regeneration among *A. thaliana* accessions depended on the expression of a gene encoding a TRX that directly modulates ROS homeostasis [62]. Despite differences in wound responses between animals and plants, ROS is a common signal in both systems [63]. In cancer cells, an increase in ROS stabilizes hypoxia-inducible factor-1α (HIF1α), which enhances glucose catabolism and allows for tumor progression [64]. Furthermore, iPSC reprogramming is enhanced under hypoxic conditions through ROS-mediated activation of HIF1α [65]. A similar hypoxia-mediated mechanism may be present in plants, but negative regulators of ERF-VII TFs may be targeted instead [66]. However, a direct link among ROS production, metabolic reprogramming, and wound-induced organ regeneration in tomato hypocotyl explants awaits experimental confirmation.

## 4. Materials and Methods

### 4.1. Plant Material and Growth Conditions

Seedlings of the tomato cultivar “Micro-Tom” were grown in vitro as described elsewhere [12]. Moneymaker (MM), *CDF3 RNAi #1.1*, *35S::CDF3 #5.2* and *35S::CDF #11.2* lines have been described elsewhere [67]. For LR emergence, 3–4 mm of the root tips in 5-day-old seedlings were excised with a sharp scalpel, and the seedlings were grown for another 2 days; newly emerged LRs were then counted. Hypocotyl explants were obtained after removing the whole root system (2–3 mm above the hypocotyl–root junction) and the shoot apex (just below the cotyledons’ petiole) with a sharp scalpel, at the 100–101 growth stages [68] (0 dae). Hypocotyl explants were transferred to 65 × 150 mm (diameter × height) glass jars with 50 mL of standard growing medium [12]. ARs were periodically counted between 2 and 12 dae. Shoot regeneration stages were quantified as described previously [14]. Data were collected 31 days later (50 das). Three to four millimeters of the apical and the basal regions of the hypocotyl were collected at 0, 24, 96 and 192 hae. Three biological replicates, each consisting of 12 apical or basal fragments were obtained.

### 4.2. Tomato TF Identification and Annotation

For TF identification, ITAG4 protein sequences were retrieved from the SolGenomics database [69] and were used for prediction of TF genes at iTAK [70], PlantTFDB [71] and PlantTFCat [72] databases. The resulting TF gene list was manually curated through PubMed searches (Appendix A). Protein sequences smaller than 90 amino acids or that did not contain a DNA binding domain identified by hmmscan in HMMER [73] were discarded. The *A. thaliana* orthologs were identified using a reciprocal best BLAST hit strategy (RBBH) with ProteinOrtho [74] using the Araport 11 [75] and tomato ITAG4 proteomes. For auxin-related genes, we used the annotation from [14].

### 4.3. RNA-Seq Analysis and GRN Generation

RNA-Seq data were obtained from the NCBI SRA database (SRR14598206–SRR14598225) and processed as described in [13]. Briefly, genes with levels below 1 cpm in the 3 libraries were discarded for further analyses, resulting in 22492 expressed genes. DEGs were filtered using a false discovery rate (FDR) < 0.01 and log2 fold change > |1|. Expression analyses were carried out using the iDEP 0.93 web application [15]. Expression profiles were generated using STEM [16]; apical and basal TF genes with ≥0.5 cpm were excluded. For hierarchical clustering and heatmap drawing, the Morpheus web-tool was used [76]. TF genes from selected STEM profiles were used to obtain their putative target genes from the PlantRegMap database [77]. GO enrichment analyses were performed by using the ShinyGO 0.76 web tool [78] with differentially expressed putative targets. GRNs of TFs and their putative targets were constructed using Cytoscape 3.2 [79], and interaction analyses were performed with the NetworkAnalyzer plugin. The annotation of peroxidases in tomato, as well as their assignment in families, was obtained from RedoxiBase [26].

### 4.4. Cell Cycle Analysis

For flow cytometry, 3–4 mm of the apical or the basal regions of the hypocotyl were collected at 0, 24, 96 and 192 hae. Nuclei were isolated using the chopping method described elsewhere [80]. The suspension was filtered over 40 µm nylon mesh, treated with RNase A (0.8 µg/mL) and stained with propidium iodide (PI; Sigma-Aldrich, St. Lois, MO, USA) (2 µg/mL) for 15 min. Nuclei were analyzed using a SH800C Cell Sorter (Sony Biotechnology, San Jose, CA, USA). Dot-Plot side scatter area (SSC-A) versus forward scatter area (FSC-A), and PI intensity area (PI-A) versus SSC-A were used to locate nuclear populations by particle side. A univariate histogram of PI (564–606 nm emission) was used to represent DNA content 2C, 4C, 8C and large ploidy nuclei on the PE-A axis. A minimum of four biological replicates including 10,000 nuclei were analyzed in each sample.

### 4.5. Cell Death and ROS Assays

For the microscopic examination of dying cells, hypocotyl explants were harvested at 0, 24, 96 and 192 hae and stained with hot acetic acid/trypan blue solution (0.5% weight/volume (*w*/*v*) in 45% acetic acid) for 3 min. After incubation for 1 h at room temperature, the hypocotyls were cleared with chloral hydrate solution (80 g chloral hydrate in 30 mL distilled water). For DAB staining, whole hypocotyl explants were harvested at 0, 24, 96 and 192 hae and immersed in a 1% *w*/*v* solution of DAB in Tris-HCl buffer, pH 4.8. After 1 h of vacuum infiltration, the samples were incubated in DAB solution overnight at room temperature. Then, the explants were cleared with chloral hydrate solution. To confirm the specific accumulation of H_2_O_2_, we incubated hypocotyl explants with 100 units/µL^−1^ catalase (Sigma-Aldrich) for 30 min before DAB staining. Thereafter, the samples were processed as indicated above.

## 5. Conclusions

Through k-means and STEM analysis of bulk RNA-Seq data, we have identified several TF genes with specific spatial and temporal expression patterns, such as *WUS*, *CUC2*, *SBRL*, *PUCHI*, and *WOX5*, to be used as reliable markers of the different stages during de novo organ formation in tomato mutants that show defects in tissue regeneration after wounding. We have identified several conserved TF modules, such as those of WIND1-ESR1/2-WOX13 and SLR/IAA14-ARF7/ARF19-SBRL, that are involved in the specification of callus and root identity during de novo shoot and root formation, respectively. In addition, candidate TFs that regulate cambial cell division in response to wounding have been identified. Our results also highlight the fact that a dynamic regulation of cell cycle and ROS could directly contribute to region-specific organ regeneration. Finally, our TF candidates need to be confirmed experimentally using all available genetic tools (i.e., known mutant alleles and CRISPR/Cas9-mediated gene editing).

## Figures and Tables

**Figure 1 ijms-23-16112-f001:**
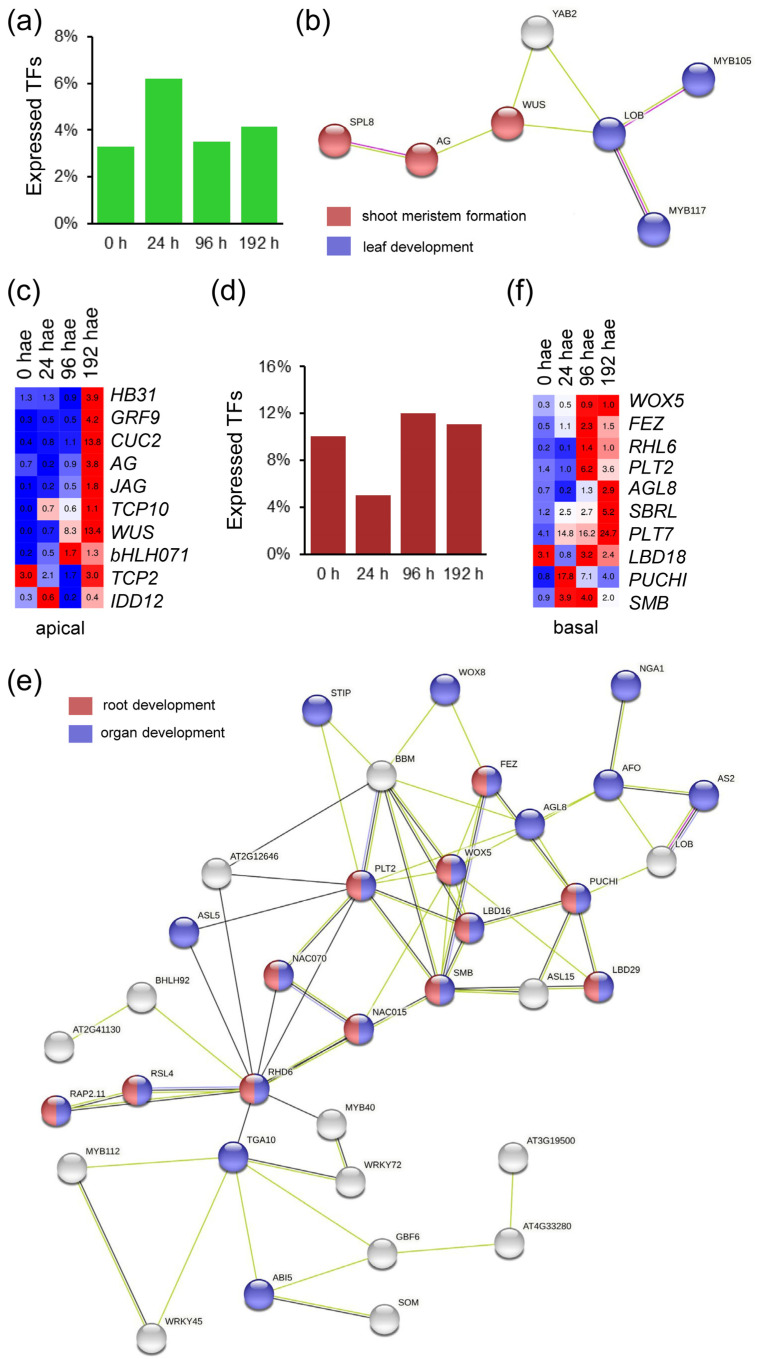
Region-specific TFs identified during de novo organ formation in tomato hypocotyl explants after wounding. (**a**) Percentage of TFs specifically expressed in the apical region of the hypocotyl over time. (**b**) STRING interaction network of *A. thaliana* orthologs of tomato TFs expressed specifically in the apical region of the hypocotyl. (**c**) Heatmap of selected TFs expressed in the apical region of the hypocotyl explants. (**d**) Percentage of TFs specifically expressed in the basal region of the hypocotyl over time. (**e**) STRING interaction network of *A. thaliana* orthologs of tomato TFs expressed specifically in the basal region of the hypocotyl. (**f**) Heatmap of selected TF expressed in the basal region of the hypocotyl explants. Blue/white/red in c and f indicate normalized counts per million reads mapped (cpm) values in each row, where red indicates the highest abundance of transcripts. Gene annotations are found in Appendix A.

**Figure 2 ijms-23-16112-f002:**
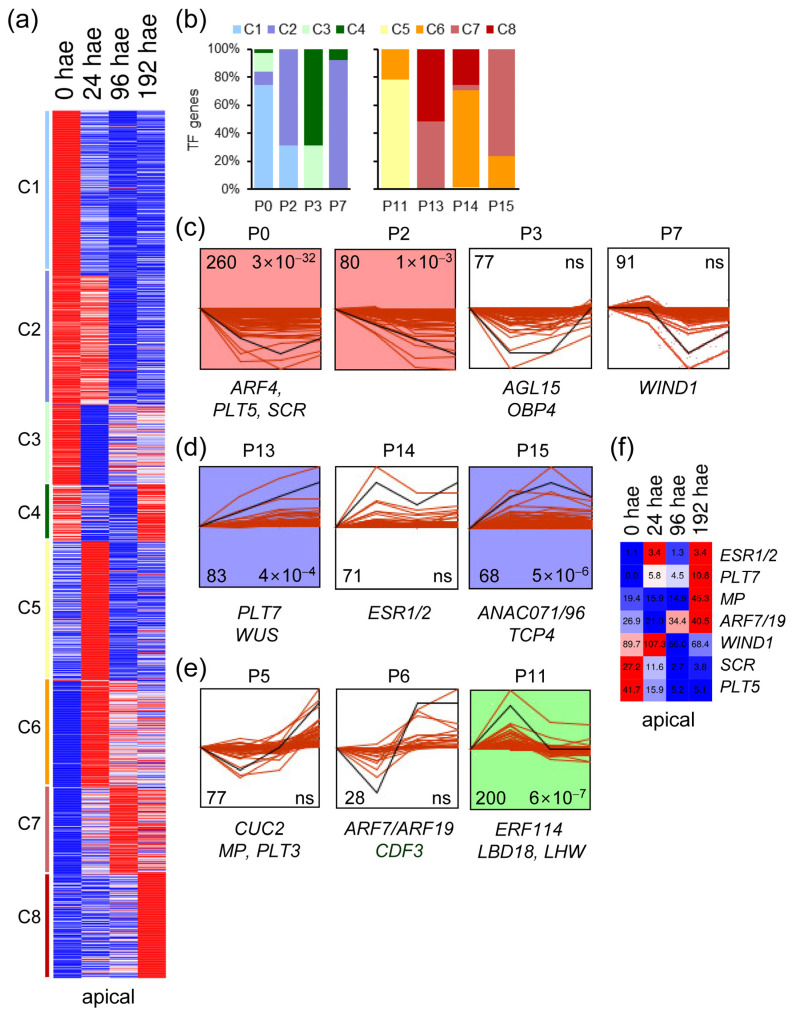
Expression profiles of TF genes during de novo shoot formation. (**a**) K-means analysis of the different TFs expressed in the apical region of the hypocotyl over time. (**b**) Percentage of k-means clusters associated with each expression profile. (**c**–**e**) Selected expression profiles of downregulated TF genes identified in STEM. Some TFs whose *A. thaliana* orthologs are known to be involved in regeneration are indicated. The numbers in the left corner indicate the number of TF genes in each expression profile, and the numbers in the right corner indicate the *p* value; ns: not significant *p* value. (**f**) Heatmap of selected TF genes with differential expression in the apical region of the hypocotyl. Blue/white/red in (**a**,**f**) indicate normalized cpm values in each row, where red indicates the highest abundance of transcripts. Gene annotations are found in Appendix A.

**Figure 3 ijms-23-16112-f003:**
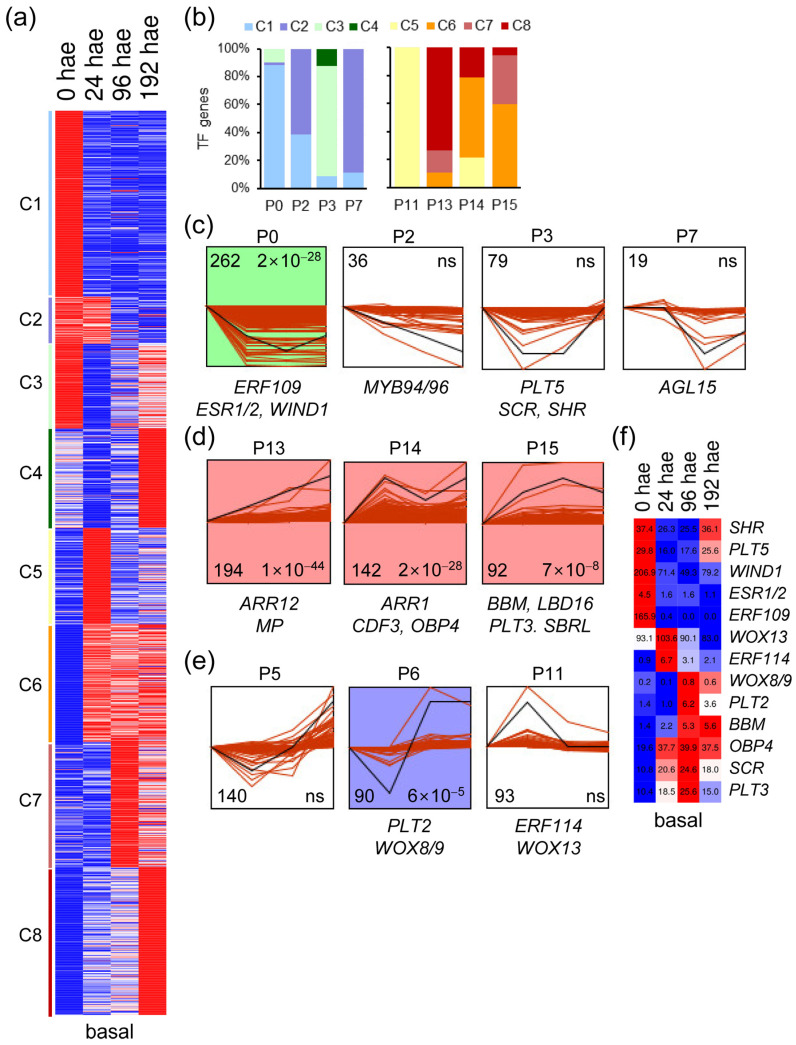
Expression profiles of TF genes during de novo AR formation. (**a**) K-means analysis of the different TFs expressed in the basal region of the hypocotyl over time. (**b**) Percentage of k-means clusters associated with each expression profile. (**c**–**e**) Selected expression profiles of downregulated TF genes identified in STEM. Some TFs whose *A. thaliana* orthologs are known to be involved in regeneration are indicated. The numbers in the left corner indicate the number of TF genes in each expression profile, and the numbers in the right corner indicate the *p* value; ns: not significant *p* value. (**f**) Heatmap of selected TF genes with differential expression in the basal region of the hypocotyl. Blue/white/red in (**a**,**f**) indicate normalized cpm values in each row, where red indicates the highest abundance of transcripts. Gene annotations are found in Appendix A.

**Figure 4 ijms-23-16112-f004:**
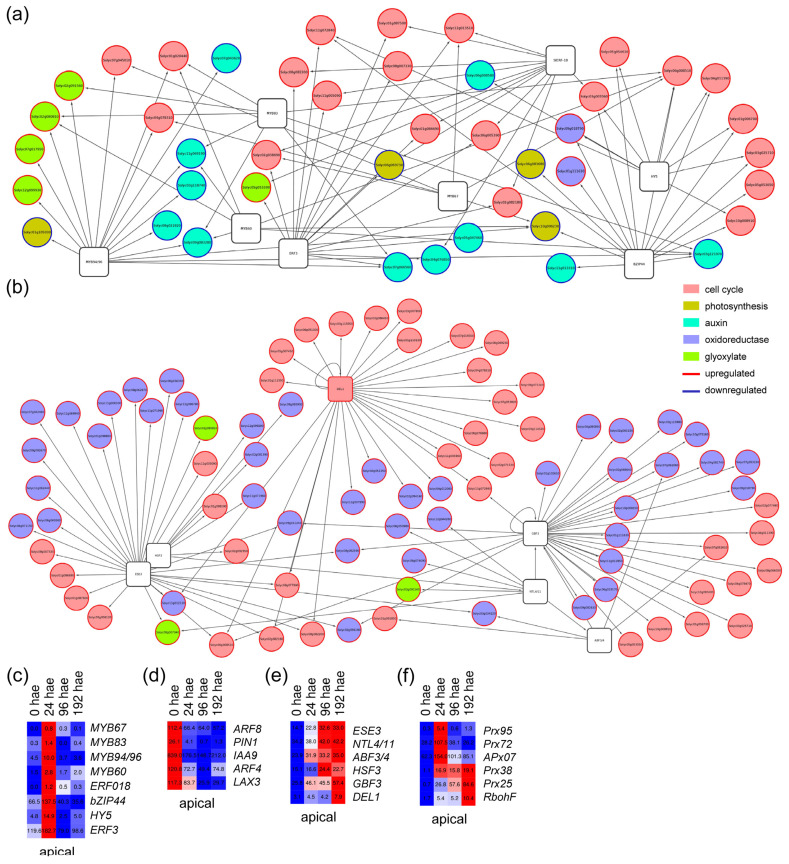
GRNs for putative TF genes and their deregulated targets associated with P11 and P13 profiles in the apical region of the hypocotyl. (**a**,**b**) Selected GRNs from P11 (**a**) and P13 (**b**) profiles. (**c**–**f**) Hierarchical clustering over time of selected TF genes (**c**,**e**) and some of their putative targets (**d**,**f**) from P11 (**c**,**d**) and P13 (**e**,**f**) profiles. Blue/white/red in (**c**–**f**) indicate normalized cpm values in each row, where red indicates the highest abundance of transcripts. Gene annotations are found in Appendix A.

**Figure 5 ijms-23-16112-f005:**
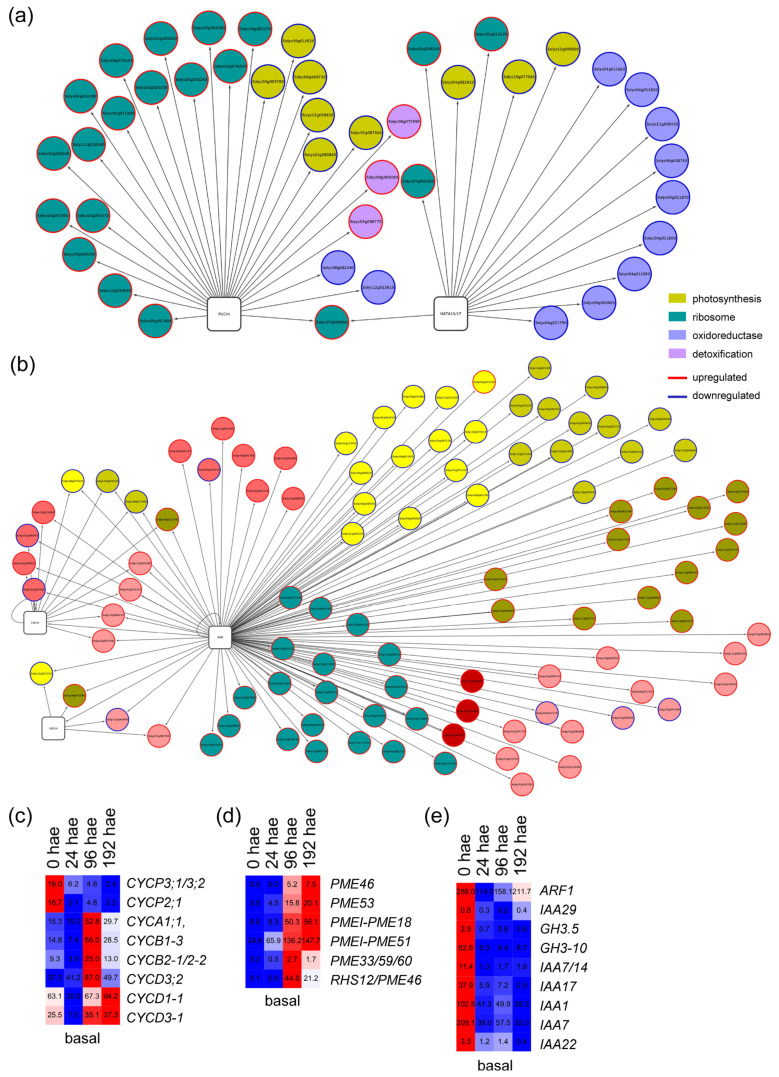
GRNs for putative TF genes and their deregulated targets associated with P11 and P15 profiles in the basal region of the hypocotyl. (**a**,**b**) Selected GRNs from P11 (**a**) and P15 (**b**) profiles. (**c**–**e**) Hierarchical clustering over time of selected putative targets from P11 (**c**,**d**) and P15 (**e**) profiles. Blue/white/red indicate in (**c**–**e**) normalized cpm values in each row, where red indicates the highest abundance of transcripts. Gene annotations are found in Appendix A.

**Figure 6 ijms-23-16112-f006:**
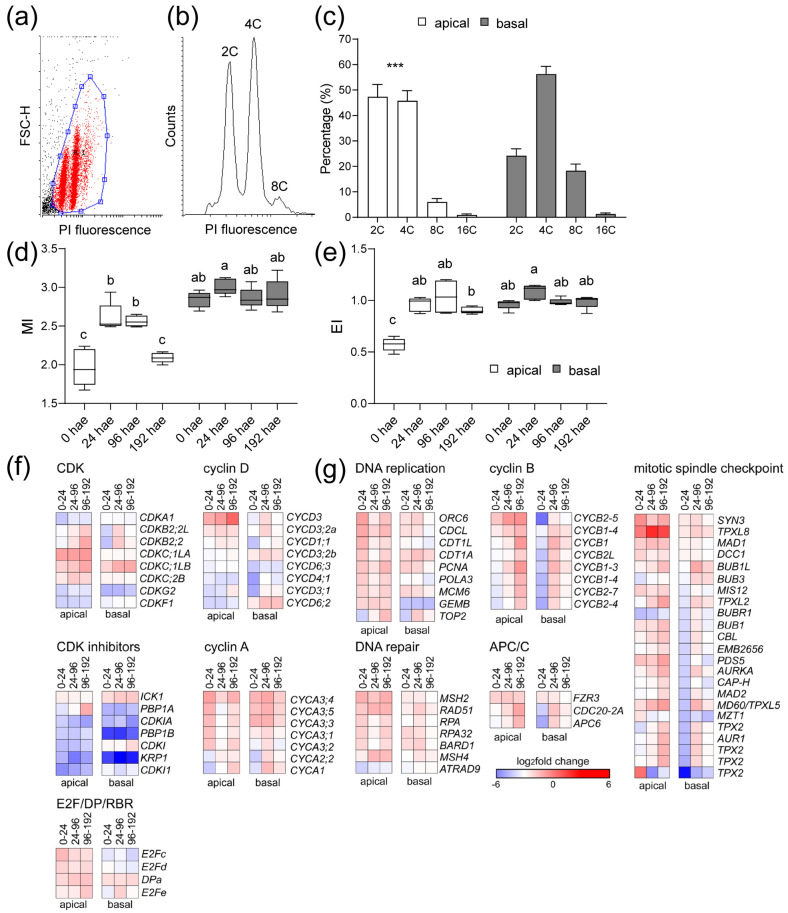
Regulation of the cell cycle during de novo organ formation. (**a**–**e**) Ploidy analysis by flow cytometry. (**a**) Bivariate scatter plot of forward scatter height (FSC-H) versus propidium iodide (PI) fluorescence. (**b**) Histogram of the DNA content of a representative sample from the apical region at 0 hae. (**c**) Proportion of nuclei with different DNA content in the apical (white bars) or the basal (black bars) region at 0 hae; asterisks indicate significant differences (*p* value < 0.001) between the regions studied. (**d**–**e**) Mitotic index (MI; (**d**)) and endoreduplication index (EI; (**e**)) over time; letters in (**d**,**e**) indicate significant differences (Fisher’s least significant difference (LSD); *p* value < 0.001) between samples. (**f**,**g**) Expression profile of DEG involved in cell cycle regulation. A set of 172 putative core cell cycle genes involved in the different phase transitions were studied (Appendix A). Blue/white/red in (**f**,**g**) indicate normalized cpm values in each row, where red indicates the highest abundance of transcripts. Gene annotations are found in Appendix A.

**Figure 7 ijms-23-16112-f007:**
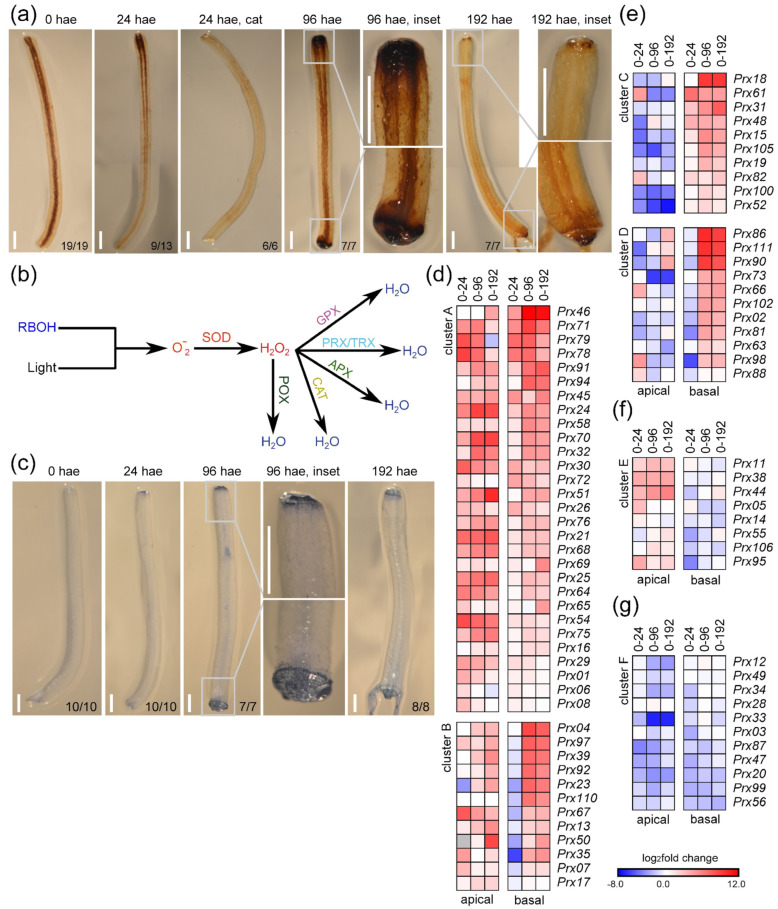
Regulation of ROS production during de novo organ formation. (**a**) H_2_O_2_ accumulation visualized by DAB staining in hypocotyl explants over time. (**b**) Proposed pathways of ROS production and detoxification with the indicated key enzyme activities: RBOH, NADPH oxidase/respiratory burst oxidase homologue; SOD, superoxide dismutase; POX, peroxidase; GPX, glutathione peroxidase; PRX/TRX, peroxiredoxins/thioredoxins; APX, ascorbate peroxidases; CAT, catalases. (**c**) Localized cell death in hypocotyl explants was visualized by trypan blue staining. (**d**–**f**) Hierarchical clustering of DEGs encoding class III peroxidases. Blue/white/red in (**d**–**g**) indicate normalized cpm values in each row, where red indicates the highest abundance of transcripts. Gene annotations are found in Appendix A. Scale bars (**a**,**c**): 3 mm.

**Figure 8 ijms-23-16112-f008:**
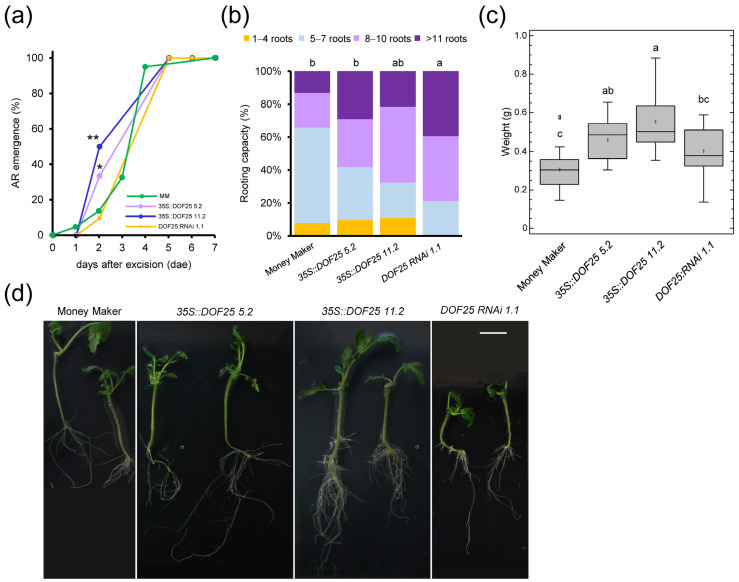
Wound-induced AR formation and shoot regeneration after apical cutting in *CDF3* tomato mutants. (**a**) AR emergence of *CDF3* overexpression lines (blue and purple), *CDF3* RNAi lines (orange) and Moneymaker (MM; green). (**b**) Rooting capacity of shoot explants at 5 dae. (**c**) Weight of the apical region of the hypocotyl after 31 dae. (**d**) Representative images of regenerated explants at 31 dae. Letters in (**b**,**c**) indicate significant differences (*p* value < 0.01) between genotypes. Asterisks in (**a**) indicate significant differences (Fisher’s LSD; *p* value < 0.05) between genotypes. Scale bars: 25 mm.

## Data Availability

Raw sequence files and read count files are publicly available in the NCBI’s BioProject PRJNA731333. Gene functional annotation is available in the Appendix A of this article. All other data that support the findings of this study are available from the corresponding author upon reasonable request.

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
