# Peer review of "Identification of Transcriptional Networks Involved in De Novo Organ Formation in Tomato Hypocotyl Explants"

_ijms, 2022, doi:10.3390/ijms232416112_

Round 1

Reviewer 1 Report

In this paper titled "Identification of Novel Transcriptional Networks Involved in de Novo Organ Formation in Tomato Hypocotyl Explants", Larriba E et al., have uncovered the gene regulatory network  involved in shoot and root regeneration.

The authors have used the key regulators identified in Arabidopsis denovo regeneration process and identified additional regulators that are co-expressed in the network.
In addition authors uncovered differential endoreduplication and ROS levels in apical vs basal regions of excised hypocotyl.
Following the TF clustering based on expression, CDF3 which has been shown to be differentially regulated in apical and basal regeneration process and the role of this gene is validated using overexpression and RNAi lines.
Minor comments:
1. The title says novel transcriptional network but most of the module explained are mostly known pathways in regeneration. please justify the title.
2. The connectivity between TF GRN and followed by ROS needs a justification. Please clearly mention if there are any TF related to ROS signaling that co-regulates ROS related pathway/network other than peroxins.
3. The authors have used CDF3 and discovered its role in adventitious root regeneration, yet the discussion about this gene in arabidopsis and if any knowledge about LR development is not explained in discussion.
4. Please mention possible interactors for CDF3 which could influence potential pathways, since in Arabidopsis CDF3 regulated gene expression is known (
https://doi.org/10.3389/fpls.2020.601558).

5. While introducing new gene/proteins in text, please expand the full name and also expand the abrreviations in graphs, legends and methods.

Author Response

Dear reviewer 1,

Thanks a lot for your helpful suggestions. Now we have changed our ms. accordingly, and included a letter with point-by-point answers (please see the attachment).

Sincerely, 

Jose Manuel

Reviewer 2 Report

Larriba and his colleagues present Novel Transcriptional Networks Involved in wound induced organ regeneration in tomato. The study highlights a functional conservation of key TF modules whose function is preserved during de novo organ formation in plants. The study could be interesting. The conclusions are supported by the data, and the manuscript is well organized. However, I have several concerns that should be addressed before publication.

Page3, Figure 1. The clear of Figure 1e should be improved.

Lowercase letters indicate significance differences in Figure 8b and Figure 8c. However, Lowercase letters “b” and “c” appears two times in the same Figure. There are confusable in the Figure, Figure title and figure note. “Figure 8b and Figure 8c” suggested changed to “Figure 8B and Figure 8C”, respectively. Similar expression is also found in Figure6.  

Moreover, the English should be improved in the whole manuscript before publication. Several corrections are given below, but these are only examples. Please check the whole manuscript.

Many sentences (eg. L 47-51; L85-90, etc/) are too long. If possible, the statement can be broken down into smaller sentences.

Page 1, L41. "One such TFs" should be changed to "One such TF".

Page 4, L102. “The 1,296 TF genes expressed in the apical region of the hypocotyl were grouped into……” suggest change to “The 1,296 TF genes expressing in the apical region of the hypocotyl were grouped into……” or “The 1,296 TF genes expressed in the apical region of the hypocotyl and were grouped into……”

Author Response

Dear reviewer 2,

Thanks a lot for your helpful suggestions. Now we have changed our ms. accordingly, and included a letter with point-by-point answers (please see the attachment).

Sincerely, 

Jose Manuel
